# Harmonic and DC Bias Hysteresis Characteristics Simulation Based on an Improved Preisach Model

**DOI:** 10.3390/ma16124385

**Published:** 2023-06-14

**Authors:** Changgeng Zhang, Haoran Li, Yakun Tian, Yongjian Li, Qingxin Yang

**Affiliations:** 1State Key Laboratory of Reliability and Inteligence of Electrical Equipment School of Electrical Engineering, Hebei University of Technology, Tianjin 300130, Chinaqxyang@tjpu.edu.cn (Q.Y.); 2Key Laboratory of Electromagnetic Field and Electrical Apparatus Reliability of Hebei Province, Hebei University of Technology, Tianjin 300130, China; 3State Grid Zhoukou Electric Power Supply Company, Zhoukou 477150, China

**Keywords:** Preisach model, DC bias, loss separation theory, oriented silicon steel sheets

## Abstract

Transformers, reactors and other electrical equipment often work under harmonics and DC-bias working conditions. It is necessary to quickly and accurately simulate the hysteresis characteristics of soft magnetic materials under various excitation conditions in order to achieve accurate calculations of core loss and the optimal design of electrical equipment. Based on Preisach hysteresis model, a parameter identification method for asymmetric hysteresis loop simulation is designed and applied to the simulation of hysteresis characteristics under bias conditions of oriented silicon steel sheets. In this paper, the limiting hysteresis loops of oriented silicon steel sheets are obtained through experiments under different working conditions. The first-order reversal curves(FORCs) with asymmetric characteristics is generated numerically, and then the Everett function is established under different DC bias conditions. The hysteresis characteristics of the oriented silicon steel sheets under harmonics and DC bias are simulated by improving FORCs identification method of the Preisach model. By comparing the results of simulation and experiment, the effectiveness of the proposed method is verified, so as to provide an important reference for material production and application.

## 1. Introduction

The concept of the Preisach hysteresis model was proposed by the German physicist F. Preisach in 1935. After improved by several researchers, the classical Preisach model with mathematical expressions was later widely used [1,2,3]. The Preisach model can be classified into continuous models and discrete models. The former needs to be realized by identifying all parameters in the distribution function which identification process is complex and computationally expensive [4,5,6,7,8], the latter requires the identification of the Everett function matrix by first order reversal curves (FORCs) [9]. However, in the actual measurement process, due to the asymmetric excitation, the DC bias component is brought, and the measurement result has a large deviation. In order to avoid the large error caused by the measurement, Naidu converted the Preisach distribution function into a form that only requires the limiting hysteresis loop to obtain the FORCs and the Everett function [10]. The classical Preisach model takes the magnetic field strength as the independent variable and the magnetic flux density as the dependent variable, it is not easy to apply in finite element analysis. Dlala deduces the specific method of determining the FORCs according to the limiting hysteresis loop and the reversal points for the inverse Preisach model with the magnetic flux density as the input and the magnetic field strength as the output. The Everett function of magnetic flux density is obtained by interpolating the FORCs [11].

Hans Hauser proposed the energetic hysteresis model based on energy balance and magnetic domain statistics theory in 1994 [12,13,14]. The model employs an energy equation to describe the hysteresis phenomenon in ferromagnetic materials, in which the magnetic energy of the crystal is decomposed into an irreversible and a reversible part, and uses a probability function to represent the interaction of domain wall motion and stray fields at the pinning center. The parameters of magnetic characteristics are related to spontaneous magnetization, magnetocrystalline anisotropy, magnetostriction, and microstructure, and can account for the effects of stress, temperature, and magnetization direction on hysteresis. In addition, compared with the J-A model, the energetic model also has the advantages of faster solution rate and fewer parameters.

In addition to the above-mentioned commonly used scalar hysteresis models, there are also some vector hysteresis models, such as the Enokizono and Soda (E&S) model [15]. The vector hysteresis models can be constructed by superimposing the scalar models in different directions, such as the vector Preisach model [16], the vector J-A model [17].

Silicon steels have been widely made as the core of transformers and motors. However, with the development of power systems, the working environment of power equipmentalso becomes more complex, especially at high-frequency, high temperature, stress, non-sinusoidal excitation. The commonly-used hysteresis models and loss calculation methods are difficult to simulate magnetic properties under complex conditions. Therefore, it is necessary to establish accurate and efficient models suitable for various complex excitation conditions to quickly simulate the magnetic properties of materials.

## 2. Dynamic Preisach Hysteresis Model

The basic idea of the classical Preisach model is that ferromagnetic materials are composed of a large number of magnetic dipoles with hysteresis properties, and in the case of an output value *u*(*t*) at a given moment, the hysteresis characteristics of ferromagnetic materials produce the sum of the hysteresis characteristics of all magnetic dipoles. The output value of the magnetic dipole *γ_αβ_u*(*t*) is assumed to be either +1 or −1 values, *μ*(*α*,*β*) is a distribution function of *α*, *β*, also known as the Preisach function (Figure 1). The classical Prcisach model can be represented by the following equation:(1)$$f\left(t\right)=\overset{}{\underset{\alpha \geq \beta }{\iint }}\;\mu \left(\alpha ,\beta \right)\gamma_{\alpha \beta }\left(u\;\left(t\right)\right)d\alpha d\beta \;$$

Based on the domain theory and statistical principles, the loss separation theory proposed by Bertotti assumes that the total loss of soft magnetic materials is divided into hysteresis loss, classical eddy current loss and excess loss, as follows:(2)$$W=W_{hy}+W_{cl}+W_{ex}$$
where: *W* is total loss, *W_hy_* is hysteresis loss, *W_cl_* is eddy current loss, *W_ex_* is excess loss.

According to the theory of ferromagnetism, hysteresis loss is caused by the irreversibility of domain wall movement and domain rotation. The hysteresis loss of the material is not affected by the frequency of magnetic field. However, eddy current loss and excess loss are greatly affected by frequency. Thus, hysteresis loss is usually calculated at an extremely low frequency.

Eddy current loss is caused by eddy current during the alternating magnetization process. The skin effect of the magnetic field can be ignored at low frequencies, and the magnetic field inside the magnetic material is uniformly distributed. Eddy current loss can be solved according to the specific shape of magnetic materials. The single-cycle eddy current loss per unit volume *W_cl_* and the corresponding loss power per unit volume *P_cl_* can be expressed as:(3)$$W_{cl}=\frac{\sigma d^{2}}{12}\int_{0}^{1/f}{\left(\frac{dB}{dt}\right)}^{2}dt$$
(4)$$P_{cl}=\frac{\sigma d^{2}}{12}{\left(\frac{dB}{dt}\right)}^{2}$$
where: *σ* is the electrical conductivity of the material, in S/m, *d* is the thickness of a single, in m, *f* is the magnetization frequency, in Hz.

Excess loss is caused by the dynamic rotation and translation of the domain inside the material under the applied magnetic field, which is caused by the microscopic eddy current. The expressions of the single-cycle excess loss per unit volume *W_ex_* and the loss power *P_ex_* per unit volume are obtained as:(5)$$W_{ex}=\sqrt{\sigma SGV_{0}}\int_{0}^{1/f}{\left|\frac{dB}{dt}\right|}^{1.5}dt$$
(6)$$P_{ex}=\sqrt{\sigma SGV_{0}}{\left|\frac{dB}{dt}\right|}^{1.5}$$
where: *S* is the cross-sectional area of the magnetic material, in m^2^; *G* is a dimensionless coefficient (*G* = 0.1375); *V*_0_ is a statistical parameter characterizing the local magnetic field distribution.

The total loss *W* can be expressed as:(7)$$\begin{array}{cc} W & =W_{hy}+W_{cl}+W_{ex}\\ & =\int_{0}^{T}H_{hy}(B)\frac{dB}{dt}dt+\int_{0}^{T}\frac{dW_{cl}}{dt}dt+\int_{0}^{T}\frac{dW_{ex}}{dt}dt\\ & =\oint_{l}\left[H_{hy}(B)+H_{cl}(B\right)+H_{ex}(B)]dB \end{array}$$
where: *H_hy_*, *H_cl_* and *H_ex_* are magnetic field strength components corresponding to hysteresis loss, eddy current loss and excess loss, respectively; *l* is the *B-H* closed path in a single period.

*H_cl_* and *H*_ex_ can be determined as follows:
(8)$$H_{cl}(B)=\frac{dW_{cl}}{dt}/\frac{dB}{dt}$$
(9)$$H_{ex}(B)=\frac{dW_{ex}}{dt}/\frac{dB}{dt}$$

Combining (3) and (8), (5) and (9), we can obtain:(10)$$H_{cl}(B)=\frac{\sigma d^{2}}{12}\frac{dB}{dt}$$
(11)$$H_{ex}=\sqrt{\sigma SGV_{0}}\lambda {\left|\frac{dB}{dt}\right|}^{0.5}$$
where: *λ* = sign(*dB/dt*) = ±1. The increase of magnetic flux density is denoted by λ = 1, while the decrease of magnetic flux density is denoted by λ = −1.

A dynamic Preisach hysteresis model with magnetic flux density *B* as the independent variable can be obtained:(12)$$\begin{array}{cc} H(B) & =H_{hy}(B)+H_{cl}(B)+H_{ex}(B)\\ & =H_{hy}(B)+\frac{\sigma d^{2}}{12}\frac{dB}{dt}+\sqrt{\sigma SGV_{0}}\lambda {\left|\frac{dB}{dt}\right|}^{0.5} \end{array}$$

When the magnetic field excitation is sinusoidal, the peak value of the magnetic flux density is *B_p_*, eddy current loss *W_cl_*(*B_p_*) and excess loss *W_ex_*(*B_p_*) can be obtained by integrating (3) and (5), respectively:(13)$$W_{cl}(B_{p})=\frac{{\left(\pi d\right)}^{2}\sigma }{6}B_{p}{}^{2}f$$
(14)$$W_{ex}(B_{p})=8.76\sqrt{\sigma SGV_{0}}B_{p}{}^{1.5}f^{0.5}$$

Hysteresis loss *W_hy_* is not affected by frequency *f*. The total loss minus eddy current loss (*W* − *W_cl_*) is represented by:(15)$$W-W_{cl}=W_{hy}+8.76\sqrt{\sigma SGV_{0}}B_{p}{}^{1.5}f^{0.5}$$

It can be seen that (*W* − *W_cl_*) is proportional to the square root of the frequency. The intersection of the corresponding function and the ordinate axis is the hysteresis loss, and the slope *k* is 8.76(*σSGV*_0_)^0.5^$B_{1.5_{p}}$. Therefore, the slope *k* can be obtained from the total loss measured by the experiment and the calculated eddy current loss, and then the statistical parameter *V*_0_ can be obtained.

In this paper, the magnetic loss of silicon steel sheets under sinusoidal excitation at different frequencies from 5 to 500 Hz is measured. Then, the relationship between (*W* − *W_cl_*) and the square root function of frequency is obtained by calculating eddy current loss as shown in Figure 2.

The solution of the statistical parameter *V*_0_ under any peak value of magnetic flux density is interpolated at discrete points, as shown in Figure 3.

## 3. Methodology

The accuracy of electromagnetic calculations depends not only on the algorithm of electromagnetic simulation, but also on the accurate characterization of the magnetic properties of materials. The magnetic properties of materials measured under standard conditions are not suitable for non-standard conditions. Therefore, it is necessary to measure the magnetic properties under non-standard conditions. In this paper, a single sheet tester (SST, 500 × 500 mm) is used to measure the magnetic properties of silicon steel sheets B27R090 under different excitation conditions. Figure 4 shows a magnetic measurement system, including SST, power supply, feedback control system and PC.

A specific waveform signal is input to the power supply using the corresponding software of the measurement system, and the corresponding excitation of the primary side coil is applied after amplification by the power amplifier. According to the Ampere circuital law, the expression for the magnetic field strength *H* is
(16)$$H(t)=\frac{N_{1}(i_{dc}+i_{ac}(t))}{l_{m}}$$
where: *N*_1_ is the number of turns of the primary side coil, *l_m_* is the equivalent magnetic circuit length, *i_dc_* is the DC excitation current; *i_ac_*(*t*) is the AC excitation current and it is necessary to apply an AC voltage *u*(*t*) containing harmonics to the primary side coil and the expression *u*(*t*) is as follows:(17)$$u\left(t\right)=U_{1m}\sin(2\pi ft)+U_{km}\sin(2k\pi ft+\theta_{k})$$
where: *U_1m_* is the amplitude of voltage at the fundamental frequency; *U_km_* is the *k*th harmonic amplitude of voltage; *f* is the fundamental frequency set as 50 Hz in the paper; *θ_k_* is the phase difference between the *k*th harmonic and the fundamental wave. In this paper, the harmonic content *η* is defined as:(18)$$\eta =\frac{B_{km}}{B_{1m}}$$
where: *B_km_* is the amplitude of the magnetic flux density of the *k*th harmonic, *B_1m_* is the amplitude of voltage of the fundamental frequency.

The magnetic flux density can be obtained by integrating the voltage of secondary side, and the expression of the magnetic flux density is as follows:(19)$$\begin{array}{cc} B(t) & =\frac{1}{N_{2}S}\int_{0}^{t}u_{2}(t)dt\\ & =-B_{1m}\cos(2\pi ft)-B_{km}\cos(2k\pi ft+\theta_{k}) \end{array}$$

The main parameters of the magnetic measurement system used in the measurement are shown in Table 1.

In this paper, the loss characteristics of silicon steel sheets under different bias currents are measured. The harmonic content of each harmonic is 30% of *U*_1_, and all the phase angles are set to be zero.

Figure 5 shows the loss characteristics under different bias currents and third harmonics. As the magnetic field strength *H*_dc_ increases from 0 A/m to 100 A/m, the loss also increases. However, as the magnetic field strength increases, the rate of the loss change gradually decreases.

## 4. Results

### 4.1. Fitting Results under Single Harmonic Excitation

The dynamic Preisach model established above is used to simulate the dynamic hysteresis loops of silicon steel sheets B27R090. Figure 6 shows the comparison of dynamic hysteresis loop simulation and measurement results with varying the harmonic contents and the amplitude of the fundamental magnetic flux density. As can be seen from Figure 6a, when the third harmonic content *η* is 20%, the hysteresis loop is only distorted inwardly without local hysteresis loops. However, when the third harmonic content is 40%, the hysteresis loops appear as two identical local hysteresis loops at symmetrical positions, as shown in Figure 6c. From the fitting accuracy of the hysteresis model to the dynamic hysteresis loop, when *B_p_* is 1.7 T, the simulated hysteresis loop is approximately the same as the measured result. Comparing between Figure 6b,d, it can be seen that the hysteresis model can still maintain the accuracy of the fitting results even with a local hysteresis loop in the hysteresis loop as *η* changes.

Figure 7 shows the comparison of dynamic hysteresis loop simulation and measurement results under various excitation conditions when *k* is 5 and the harmonic phase difference *θ_5_* is 0°. It can be seen from Figure 7 that when the fifth harmonic content is 20%, there are four local hysteresis loops that are independent of each other, and when the fifth harmonic content is 40%, the four local hysteresis loops become two-crossed, as shown in Figure 7c. Compared with Figure 6, it can be seen that the hysteresis loops under the excitation of the fifth harmonic with the same content of 20% appear a local hysteresis loop. It can be seen that with the increase in harmonic order, the influence of harmonics on the shape of hysteresis loop becomes more complex. From the perspective of the fitting accuracy of the hysteresis model on the dynamic hysteresis loop, though the local hysteresis loops become crossed, the hysteresis model can still simulate the hysteresis loop accurately.

The hysteresis loops and losses are simulated by the dynamic hysteresis model under various excitation conditions. Table 2 shows the comparison between the model calculation loss result *P*_cal_ and the measured loss result *P*_mea_ under each excitation condition when the harmonic order *k* = 3 and 5. It can be seen from Table 2 that the loss increase as increasing the harmonic order and the harmonic content. Under these different excitation conditions, the relative error of the model calculation loss is within 4%, which meets the accuracy requirements.

### 4.2. Simulation of Hysteresis Characteristics under AC-DC Hybrid Excitation

The fitting accuracy of the Preisach hysteresis model mainly depends on the Everett function. When the excitation contains DC component, the limiting hysteresis loop of the soft magnetic material will become asymmetrical, the classical method is no longer applicable and need to be improved.

#### Hysteresis Model under DC Bias

Figure 8 shows the quasi-static hysteresis loop with DC bias *H_dc_* = 10 A/m. The improved method generates FORCs based on the distance between each point and the ascending or descending branch, the improved method can identify the curve regardless of the change in the limiting hysteresis loop.

For any turning point G on the descending branch, the ascending branch and the distance Δ*B*_g_ to *B* = *B_Q_* are given, respectively:(20)$$\Delta H_{g}=H_{a}\left(B_{G}\right)-H_{d}\left(B_{G}\right)$$
(21)$$\Delta B_{g}=B_{Q}-B_{G}$$

For any point P on the GPQ, the distance to the ascending branch is Δ*H*, then the magnetic field strength *H_P_* at point *P* is:(22)$$H_{P}=H_{a}\left(B_{P}\right)-\Delta H$$

During the movement of point P from point G to point Q, Δ*H* decreases continuously from Δ*H*_g_ to zero, which can be expressed as a function of Δ*H_g_* and the limiting hysteresis loop width Δ*H_P_ = H_a_*(*B_p_*) *− H_d_*(*B_p_*), and finally obtain *H_P_* as:(23)$$H_{P}=H_{a}\left(B_{P}\right)-\Psi \left(\Delta H_{g},\Delta H_{P}\right)$$

According to the above improved method, the FORCs of B27R090 under the excitation of AC magnetic flux density peak value of *B_acp_* = 1.9 T, AC frequency *f* = 5 Hz, and DC bias magnetization of *H_dc_* = 50 A/m is obtained, as shown in Figure 9. The Everett functions based on the single curve of the descending branch and the ascending branch are obtained by solving, respectively, as shown in Figure 10.

The static hysteresis loop at *H_dc_* = 50 A/m is fitted by the improved method, and the peak values of the AC magnetic flux density are 1.0 T and 1.5 T, respectively, and the results are shown in Figure 11. It can be seen that the model fitting results are approximately the same as the measurement results.

When the DC bias *H_dc_* ranges from 10 A/m to 70 A/m, the difference between the total loss and eddy current loss (*W* − *W_cl_*) is calculated, and *V*_0_ is obtained by solving the slope of the linear regression function, as shown in Figure 12.

When the peak value of the AC magnetic flux density *B_acp_* is less than 1.0 T, *V*_0_ increases with the increase in *H_dc_* at the same *B_acp_*, and when *B_acp_* is greater than 1.0 T, the *V*_0_ at different bias currents almost keep constant. Therefore, *V*_0_ is determined by both *B_acp_* and *H_dc_* under AC-DC mixed excitation.

*V*_0_(*B_acp_*, *H_dc_*) is fitted by cubic polynomial, and the expression *V*_0_(*B_acp_*, *H_dc_*) is shown in (20), a1~a7 are the fitting parameters, which are 0.01534, 0.0005907, −0.01629, −0.00168, −0.02066, 0.001131 and 0.04335, respectively, and the fitting results are shown in Figure 13.
(24)$$\begin{array}{cc} V_{0}(B_{acp},H_{dc}) & =a_{1}B_{acp}^{3}+a_{2}B_{acp}^{2}H_{dc}+a_{3}B_{acp}^{2}\\ & +a_{4}B_{acp}H_{dc}+a_{5}B_{acp}+a_{6}H_{dc}+a_{7} \end{array}$$

Compared with the improved method, loss separation theory and *V*_0_ fitting results under DC bias, the dynamic hysteresis loop under AC/DC hybrid excitation is simulated, and the results are shown in Figure 14, the peak value of the AC magnetic flux density *B_acp_* = 1.7 T in each excitation condition. Compared with no bias current, the dynamic hysteresis loop under AC/DC hybrid excitation is mainly distorted in the upper and lower parts, and the middle part is still greatly affected by harmonics. The comparison between the loss calculation result *P_cal_* and the measurement result *P_mea_* is listed in Table 3. It can be seen that the improved model is more suitable for simulating the dynamic hysteresis loop under the AC-DC mixed excitation. Compared with the hysteresis loop, it can be seen that the DC bias leads to the magnetization saturation only in the bias direction, which increases the magnetic loss of soft magnetic materials.

## 5. Conclusions

Based on the asymmetric limiting hysteresis loop, the first-order reversal curve under bias magnetic field is generated, and the parameter identification of the Preisach hysteresis model is realized, which can simulate the harmonic bias hysteresis characteristics more accurately.Considering the influence of DC bias magnetic field and AC flux density peak on excess loss, the function formula of relevant parameters in excess loss is constructed, and the parameters are extracted based on the experimental results, so as to realize the accurate simulation of hysteresis loop under DC bias conditions.When the DC magnetic field strength increases from 0 A/m to 100 A/m, the loss also increases. However, the speed of loss change gradually decreases as the magnetic field strength increases.The improved numerical simulation method in this paper is suitable for the simulation of asymmetric hysteresis characteristics. However, due to the different types of silicon steel sheets, the influence of DC bias on total loss and different types of loss will also be different. This paper considers the hysteresis characteristics of DC bias, but not other working conditions, such as the influence of temperature and stress on the hysteresis loop that need to be studied and improved in the future.

## Figures and Tables

**Figure 1 materials-16-04385-f001:**
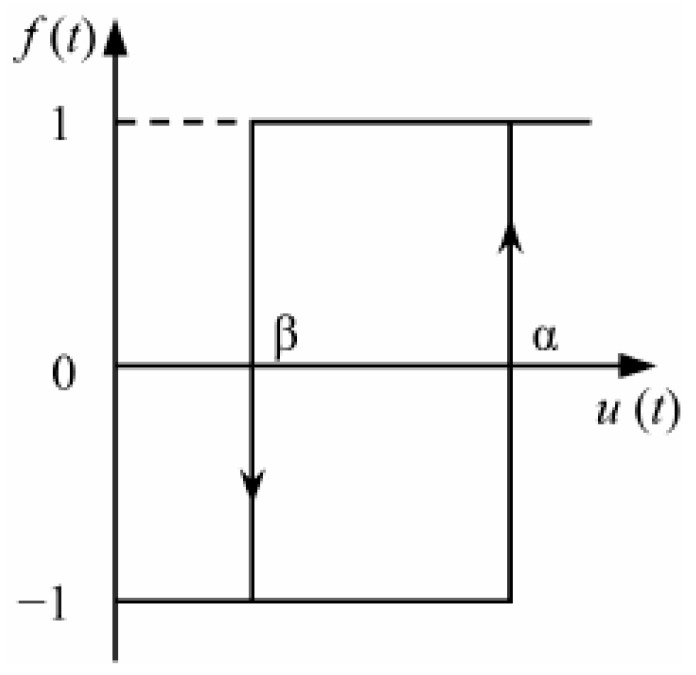
Hysteresis operator.

**Figure 2 materials-16-04385-f002:**
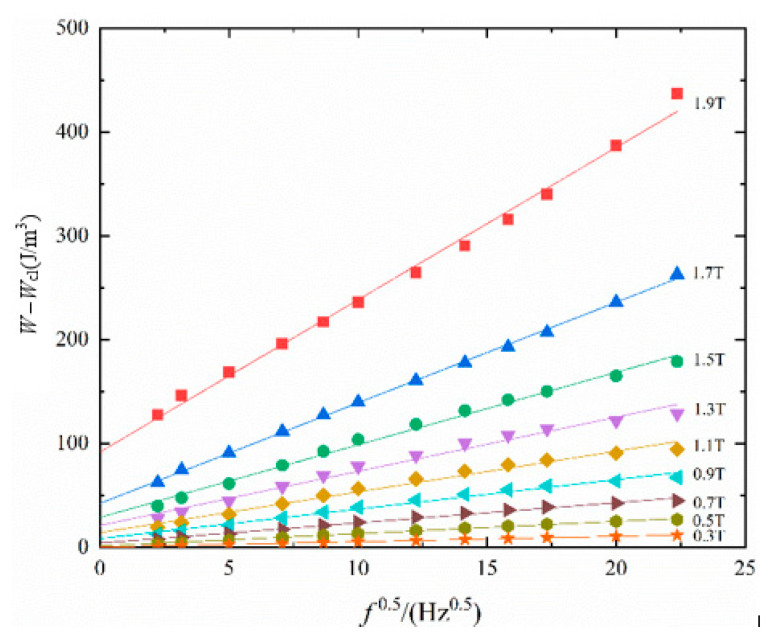
Linear regression relationship between (*W* − *W_cl_*) and *f*^0.5^ under sinusoidal excitation.

**Figure 3 materials-16-04385-f003:**
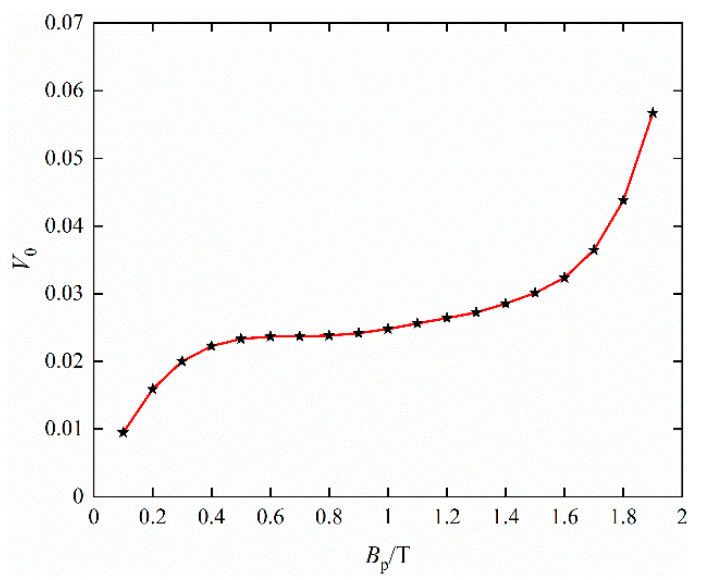
Value of *V*_0_ of different magnetic flux density under sinusoidal excitation.

**Figure 4 materials-16-04385-f004:**
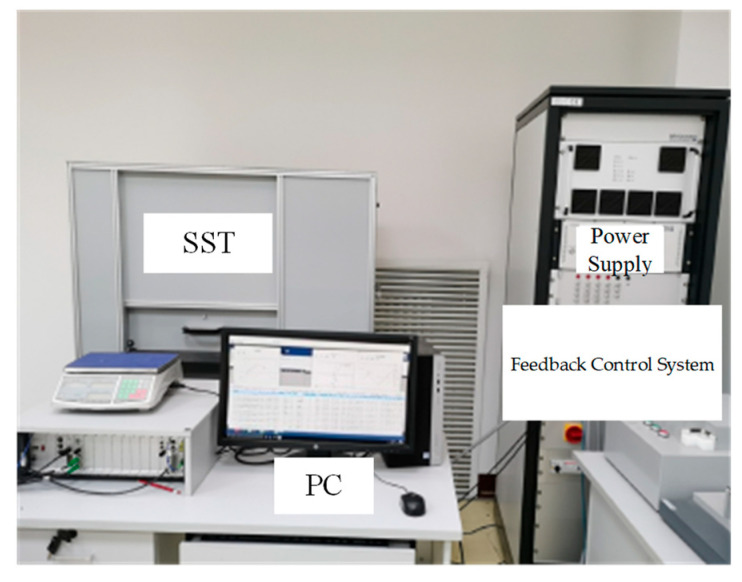
Magnetic measurement system.

**Figure 5 materials-16-04385-f005:**
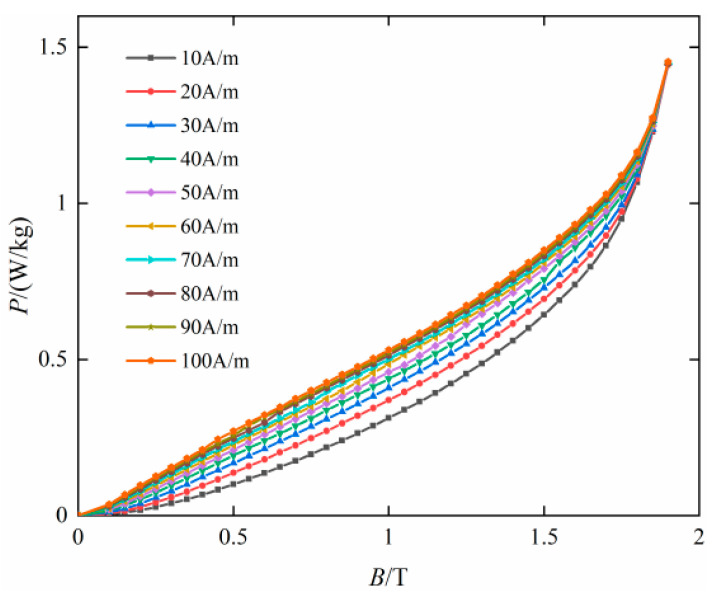
Loss characteristics under different bias currents.

**Figure 6 materials-16-04385-f006:**
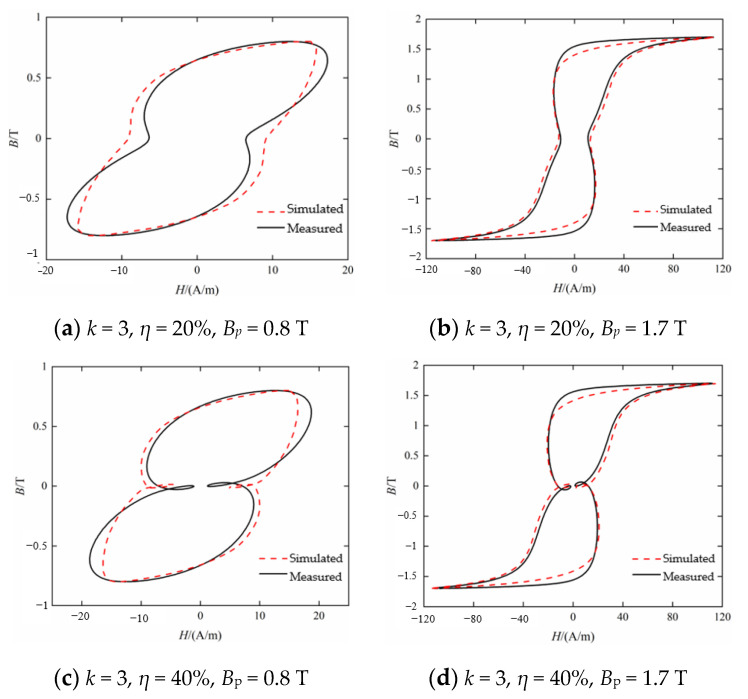
Comparison between measured and calculated hysteresis loop under *k* = 3 (*θ_3_* = 0°).

**Figure 7 materials-16-04385-f007:**
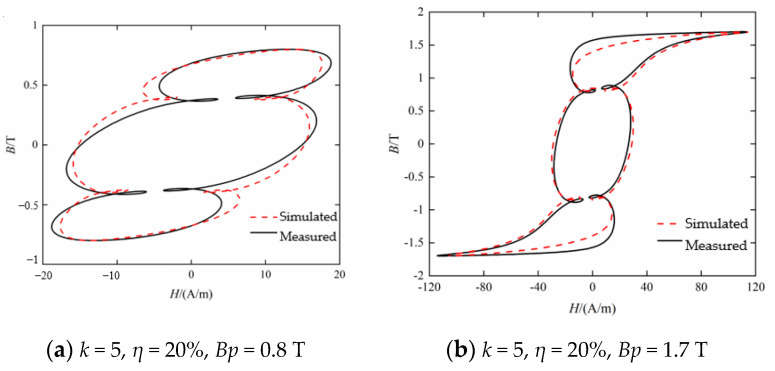
Comparison between measured and calculated hysteresis loop under *k* = 5 (*θ*_5_ = 0°).

**Figure 8 materials-16-04385-f008:**
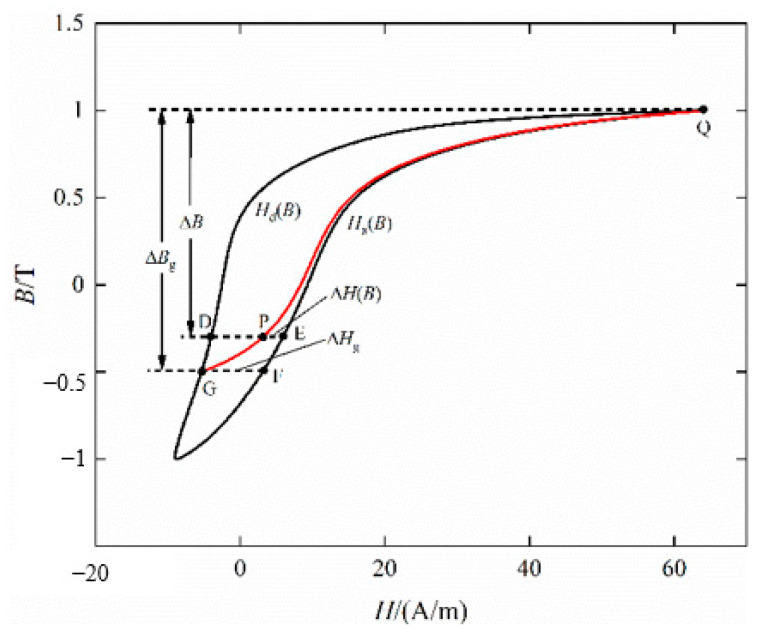
Identification of FORCs based on asymmetric limiting magnetic hysteresis loop.

**Figure 9 materials-16-04385-f009:**
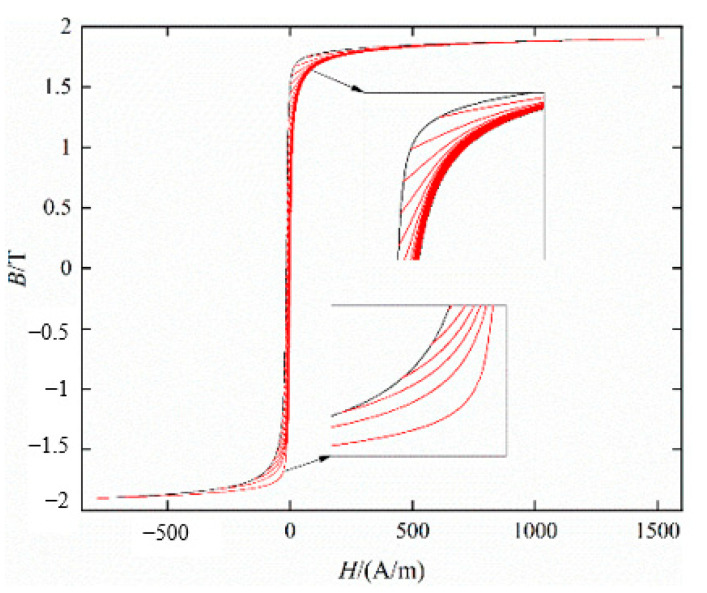
FORCs based on asymmetric limiting magnetic hysteresis loop (*H_dc_* = 50 A/m).

**Figure 10 materials-16-04385-f010:**
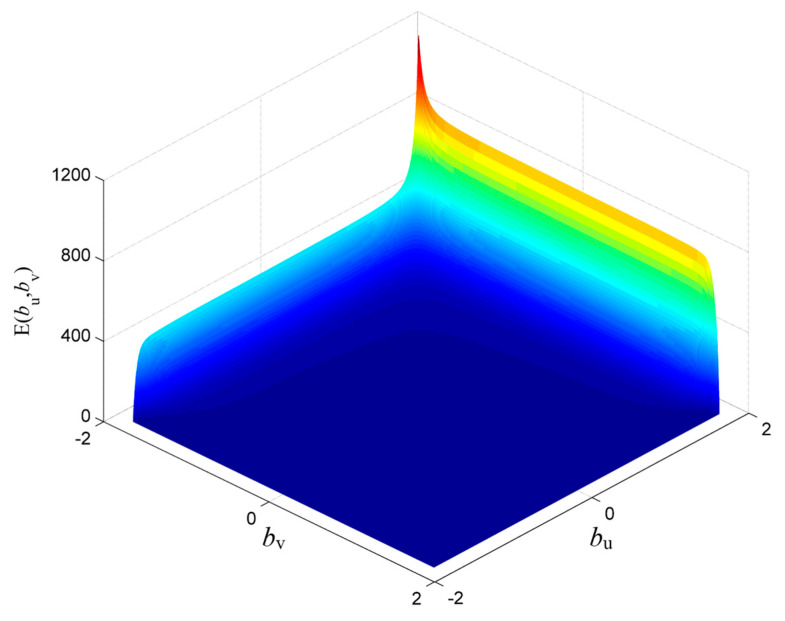
Everett function based on asymmetric limiting magnetic hysteresis loop (*H_dc_* = 50 A/m).

**Figure 11 materials-16-04385-f011:**
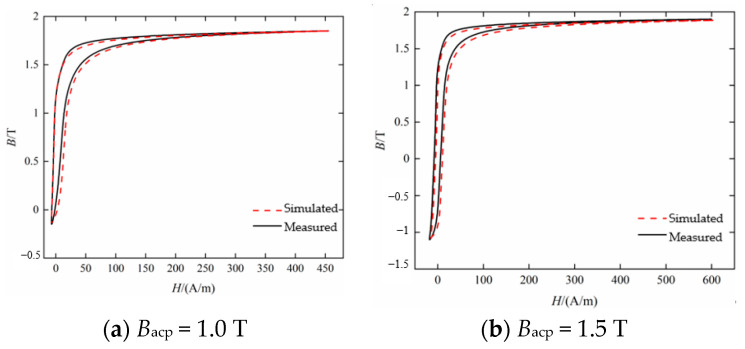
Comparison between measured and calculated static hysteresis loop under *H_dc_* = 50 A/m.

**Figure 12 materials-16-04385-f012:**
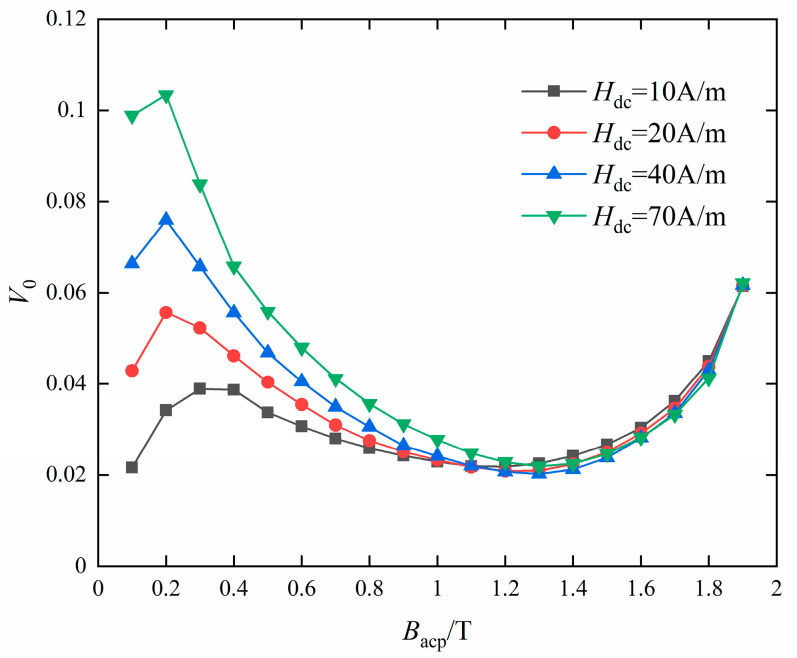
Value of *V*_0_ of different magnetic flux density under DC bias.

**Figure 13 materials-16-04385-f013:**
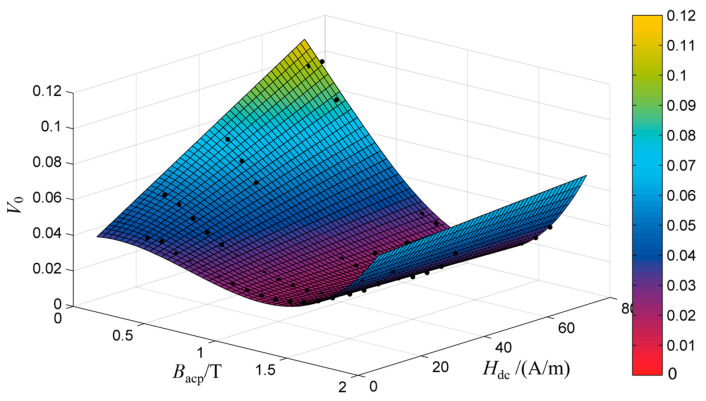
Fitting results of *V*_0_(*B_acp_*, *H_dc_*).

**Figure 14 materials-16-04385-f014:**
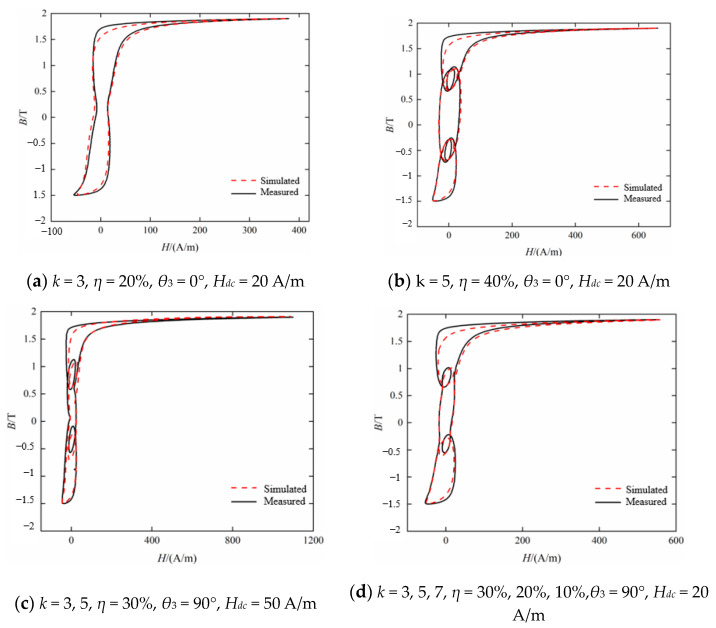
Comparison between measured and calculated dynamics.

**Table 1 materials-16-04385-t001:** The main parameters of the magnetic measurement system.

Parameter	Numeric Value
*B*/T	0.001~2
*H*/(A/m)	1~10,000
*N*_1_:*N*_2_	197:199
Equivalent magnetic circuit length/m	0.45

**Table 2 materials-16-04385-t002:** Comparison between measured and calculated hysteresis loop under *k* = 3 and 5.

Excitation	*η* (%)	*B*_p_ (T)	*P*_cal_ (W/kg)	*P*_mea_ (W/kg)	Relative Error
*k* = 3 *θ* = 0°	20	0.8	0.1856	0.1795	3.40%
	20	1.7	0.8841	0.9129	3.15%
	40	0.8	0.2087	0.2101	0.67%
	40	1.7	1.014	1.0410	2.59%
*k* = 5 *θ* = 0°	20	0.8	0.2305	0.2229	3.41%
	20	1.7	1.0658	1.0956	2.72%
	40	0.8	0.3541	0.3495	1.32%
	40	1.7	1.5893	1.6328	2.66%

**Table 3 materials-16-04385-t003:** Comparison between measured and calculated loss under AC-DC excitation (*B*_acp_ = 1.7 T).

Excitation	*θ* (°)	*H_dc_(*A/m*)*	*P*_cal_ (W/kg)	*P*_mea_ (W/kg)	*P_H_*_dc=0_ (W/kg)	Relative Error
*k* = 3 (20%)	0	20	0.9385	0.9511	0.9129	1.32%
*k* = 5 (40%)	0	20	1.6549	1.7165	1.6328	3.59%
*k* = 3, 5 (30%)	90	50	1.4510	1.5152	1.3703	4.23%
*k* = 3, 5, 7 (30%, 20%, 10%)	90	20	1.2157	1.2688	1.2117	4.19%

## Data Availability

Data cannot be used due to privacy or ethical restrictions.

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
