# Peer review of "Harmonic and DC Bias Hysteresis Characteristics Simulation Based on an Improved Preisach Model"

_materials, 2023, doi:10.3390/ma16124385_

Round 1
Reviewer 1 Report
Please refer attached file.

Author Response
I thank the experts for their comments. The comments made have been revised in the text

Reviewer 2 Report
|
1 |
Originality, adherence to the fundamentals and rules of the scientific research methodology |
|
The paper is an original scientific article that follows all scientific processes and adheres to the fundamentals of science.
|
|
|
2 |
Depth of scientific processing and understanding the subject elements |
|
The author has a thorough comprehension of scientific processing and subject aspects.
|
|
|
3 |
Scientific addition and active contribution to specialization |
|
The study will contribute to scientific understanding, and the author has made a significant contribution to this article |
|
|
|
|
|
4 |
The researcher's ability to Analyze, Conclusive and Criticize |
|
The author has a strong analytical, conclusive, and critical skill.
|
|
|
5 |
Authenticity and diversity of sources and references |
|
All references are diverse, and the sources are credible.
|
|
|
6 |
Accuracy of documentation and commitment of scientific originality |
|
The correctness of documentation and the greatest degree of regard for scientific originality have been prioritized by the author. |
|
|
7 |
Language Clarity |
|
The author has taken care to ensure that the language is clear and that the grammar is acceptable. |
|
Comments for Authors
Abstract:
Put a definition of abbreviations in the abstract like
J-A model. The full form of DC is direct current.
Introduction
It is suggested that further quoted articles can be added to make the introduction more precise and substantial:
doi: 10.1109/MIAS.2016.2600721
doi: 10.1016/j.matpr.2020.08.3893
doi: 10.1016/j.cja.2022.03.002
Discussion is very short
Author Response

(The authors gave the same response as above.)

Reviewer 3 Report
The development of simple and accurate methods for simulating hysteresis characteristics under various excitation conditions is an important task contributing to the use of soft magnetic materials. Therefore, the present article is relevant and interesting and the main conclusions were confirmed by mathematical and simulation analyses. My comments are listed below.
1. The graphical presentation of the article should be improved. For example, in some figures (especially Аigures 7 and 10), both the axes' labels and tick labels are poorly visible.
2. Section 4 is titled "Discussion", although it is more in line with the "Conclusion" section. It should be corrected.
Author Response

(The authors gave the same response as above.)

Round 2
Reviewer 1 Report
Please see attached file.

Author Response
1)I'm sorry I didn't annotate the revised part in the paper, the revised part has been marked in the revised paper.
2)The conclusions and summary sections have been revised
3)The limitations of the model are explained in the conclusion section
4)Added methodology section. The loss characteristics of the material were analyzed
5)Added illustrations of Figures 10 and 13
